# Feasibility Study of an Internet-Based Platform for Tele-Neuropsychological Assessment of Elderly in Remote Areas

**DOI:** 10.3390/diagnostics12040925

**Published:** 2022-04-07

**Authors:** Radia Zeghari, Rachid Guerchouche, Minh Tran-Duc, François Bremond, Kai Langel, Inez Ramakers, Nathalie Amiel, Maria Pascale Lemoine, Vincent Bultingaire, Valeria Manera, Philippe Robert, Alexandra König

**Affiliations:** 1Cognition Behavior Technology (CoBTeK) Lab, FRIS-Université Côte d’Azur, 06000 Nice, France; rachid.guerchouche@inria.fr (R.G.); francois.bremond@inria.fr (F.B.); valeria.manera@univ-cotedazur.fr (V.M.); philippe.robert@univ-cotedazur.fr (P.R.); alexandra.konig@inria.fr (A.K.); 2Institut National de Recherche en Informatique et en Automatique (INRIA), 06902 Valbonne, France; tran-duc.minh@inria.fr; 3Janssen Clinical Innovation (JCI), Janssen Research & Development, Division of Janssen Pharmaceutica NV, 2340 Beerse, Belgium; klangel1@its.jnj.com; 4MHeNs School for Mental Health and Neuroscience, Maastricht University, 6211 LK Maastricht, The Netherlands; i.ramakers@maastrichtuniversity.nl; 5Consultation Mémoire, Centre Hospitalier Digne-les-Bains, 04000 Digne-les-Bains, France; namiel@ch-digne.fr (N.A.); mplemoine@ch-digne.fr (M.P.L.); vbultingaire@ch-digne.fr (V.B.)

**Keywords:** telemedicine, cognitive testing, dementia, teleneuropsychology, neuropsychology

## Abstract

Today, in rural isolated areas or so-called ‘medical deserts’, access to diagnosis and care is very limited. With the current pandemic crisis, now even more than ever, telemedicine platforms are gradually more employed for remote medical assessment. Only a few are tailored to comprehensive teleneuropsychological assessment of older adults. Hence, our study focuses on evaluating the feasibility of performing a remote neuropsychological assessment of older adults suffering from a cognitive complaint. 50 participants (aged 55 and older) were recruited at the local hospital of Digne-les-Bains, France. A brief neuropsychological assessment including a short clinical interview and several validated neuropsychological tests was administered in two conditions, once by Teleneuropsychology (TNP) and once by Face-to-Face (FTF) in a crossover design. Acceptability and user experience was assessed through questionnaires. Results show high agreement in most tests between the FTF and TNP conditions. The TNP was overall well accepted by the participants. However, differences in test performances were observed, which urges the need to validate TNP tests with broader samples with normative data.

## 1. Introduction

Today, developing remote solutions for clinical assessments has become more important than ever. Indeed, access to healthcare can be complicated, especially in underserved so-called “medical deserts”. In France, this is affecting approximately 5.7% of the general population [1]. Regarding specialists, the situation is even more alarming, particularly when it comes to Alzheimer’s disease (AD) and other related disorders. 900,000 people in France are affected by AD with 225,000 new cases every year. People suffering from AD are under-diagnosed, with only 50% being detected at the beginning of the disease, precisely because of limited access to specialized health professionals [2]. This problem is of even greater importance in rural areas and can lead as consequence to suboptimal symptom management, missed opportunities to provide lifestyle and other behavioral interventions, and increased caregiver burden [3]. The global COVID-19 pandemic has highlighted even more the urgent need for innovative solutions allowing for remote screening and monitoring of cognitive decline.

Administering cognitive assessment via video-teleconference (VTC) platforms may provide opportunities to mitigate some of these challenges. Nineteen studies investigating such solutions were identified in 2020 for older adults in various domains, ranging from AD and other related disorders to rehabilitation and psychiatry [4]. Several of them have reported good agreement between VTC or Tele Neuropsychological (TNP) assessment and face to face -administered neuropsychological assessment [4,5]. Furthermore, feasibility studies demonstrated good agreement for diagnosis and the ability to safely store private data. Visual, motor and verbal cognitive tests showed a good validity and strong agreement between the tests administered face to face and in VTC [5]. Verbal tests such as verbal fluency or list learning tasks were not affected by the remote modality. There seem to be more variability for tests requiring such as the MMSE (drawing on a sheet of paper) or the clock drawing test. This variability of results depends as well on age and internet access speed.

A recent meta-analysis by Marra and Colleagues updated previous work on remote neuropsychological tests validity and reliability [4,5]. The authors explained that despite the interest of teleneuropsychology, it has not been used yet by practitioners due to a lack of reimbursement from Medicare and private insurances. Another issue is the lack of correct adaptation and implementation of test material in TNP solutions.

In French, only a few neuropsychological tests have been digitized and validated with their specific designs and test material [6] such as the MoCA (Montreal Cognitive Assessment) [6] or the CANS-MCI (Computer-Administered Neuropsychological Screen) for assessing mild cognitive impairment [7]. Three main barriers to teleneuropsychology have been identified [8]. The first one is that changes in test format could impact its function as the perceptual, cognitive and/or motor complexity of a task would change as well. The second one is that variation could exist between two different devices (smartphone or tablet). The third barrier is that the interface would vary with the landscape and this variation would be unpredictable with the development of new technologies. Therefore, more studies need to be carried out in the future on digitized neuropsychological tests with adapted interfaces and settings to verify their feasibility as well as validity. In the specific case of teleneuropsychology for older adults, the APA proposed a protocol with a list of tests and the appropriate set-up (a large screen monitor or a standard–size Ipad, a 120 min battery laptop...etc.) and telehealth platform suggestions (zoom displaying a slide or a pdf) [9] which consist mainly of usual conference systems such as Zoom or Skype. However, these remain relatively limited in functionalities and do not support most of of neuropsychological tests necessary for a more detailed assessment.

Thus, we are interested in investigating whether a short neuropsychological evaluation covering all cognitive functions is feasible to be performed remotely via VTC. For this purpose, we developed a platform entirely dedicated to remote psychological and cognitive evaluation. We included different modules for anamnesis and medical history, and test modules with cognitive tests (memory, attention, processing speed, etc.) and behavioral scales (depression, apathy, subjective functioning). With these modules, the assessment can also be used for short and long interviews with doctors (general practitioners, psychiatrists, psychologists). In a next step, we carried out a feasibility study in a Memory Clinic for the use of this TNP system specifically designed for remote cognitive evaluation.

There are several ways to perform a neuropsychological assessment remotely either by performing the tests exactly as they are administered in the clinic, by digitalizing cognitive tests, or by creating new evaluation tests adapted to this modality [10]. In our study, we did not create new evaluation tests but implemented and adapted already existing ones in the TNP system. On the one hand, we digitize certain parts of the tests as for the Mini Mental State Examination (MMSE) [11] or the Free Cued Selective Reminding Test [12] where subjects could read on the screen instead of paper items (e.g., word list, instructions “Close your eyes”). On the other hand, tests were carried out in the exact way but at a distance such as for the verbal fluency tests. The aim of our study is to verify whether the remote assessment is reliable compared to classical testing.

## 2. Materials and Methods

### 2.1. Participants and Protocol

50 patients with cognitive complaints were recruited from the local hospital in Digne-Les-Bains, a remote rural area far from big cities in France. The study was approved by the Nancy Ethics Committee (ID RCB: 2019-A01225-52) and was conducted in accordance with the Helsinki Declaration. Written, informed consent was obtained from all subjects.

### 2.2. Procedure/Protocol

A comprehensive neuropsychological assessment was administered face-to-face (FTF) and via the TNP system two weeks apart (see Table 1).

Two psychologists from Digne-Les-Bains conducted the FTF assessment and two psychologists the TNP assessment from Nice. We designed a cross-over procedure so that half of the participants experienced the FTF screening test first and the other half the TNP screening test first. All patients performed the assessment on a desktop computer in a quiet room within the hospital of Digne-les-bains. For the TNP testing, a nurse from the hospital switched the computer on, and logged participants to their session. She also provided materials for the tests (white paper and a pencil for the MMSE test). When the participants were installed and the psychologist connected to the session, the nurse left the room but stayed in another adjacent room in case of any technical issues or if the participant needed anything (water, discomfort, volume adjustments). Afterwards, the psychologist from Nice could take control of the computer at distance. The participants were then asked to answer the questions and perform the assessment.

Two parallel versions of the tests were administered in the FTF and the TNP conditions to avoid learning effects for the three recall words of the MMSE, the FCSRT (lists “Hareng”/“Sardine”), and Fluency tasks (Semantic category: Fruits/Animals; Letter: P/R). Participants of the study were asked to complete a questionnaire on their experience and acceptability of the TNP assessment. A few tests (Cookie Theft Picture [17]), Digit Span Onwards and Backwards, positive and negative event recall [18], door test [19] were also included for the TNP assessment but not included in the analysis since they were not performed face-to-face in Digne. These tests enabled us to assess remotely the participants’ language, working memory, visual episodic memory, but also to perform completely the FCSRT which requires a pause of 20 min between the learning phase and the differed recall.

### 2.3. The Tele-Neuropsychology System: PsyTime

The developed TNP tool PsyTime is a web-based platform developed specifically for assessing and monitoring remotely patients with a cognitive complaint. This web platform allows an easy and direct connection between a clinician (e.g., neuropsychologist, psychiatrist, geriatrician) and their patient or study participant. Both connect to the web platform using their respective identifiers and passwords. The patient’s interface design is very clear and simple allowing its use by individuals who are not familiar or not able to use computers and internet browsing. The clinician’s interface is more elaborated to allow several functionalities (see Figure 1). The platform allows to administer cognitive tests, affective scales (e.g., depression and apathy scales), interviews with the ability to comment and annotate easily for assessors.

### 2.4. Acceptability Evaluation

All participants were asked to complete a questionnaire (see Appendix A) on the acceptance of the TNP system based on the ‘System Usability Scale’ [20]. The scale assesses the user experience on a Likert scale ranging from 1 (I completely disagree) to 7 (I completely agree), including items on the overall experience, if participants were satisfied, if they want to repeat the experience, and clarity of instructions as well as what type of method is preferred and why, and what could be improved.

### 2.5. Data Analysis

ICC estimates and their 95% confident intervals were calculated using SPSS statistical package version 23 (SPSS Inc., Chicago, IL, USA) based on an absolute-agreement, single measurement, 2-way mixed-effects model.

We compared the total scores of the MMSE (from 0 to 30). For the FCSRT, we compared the total recall score (0–48), the total free recall score (from 0–16), and the delayed recall score (0–16). For the naming task (LEXIS), we compared the total score (0–64). We compared reading’s duration of the three subitem for the STROOP test (color, reading, interference). For verbal fluencies, we compared the total number of words said for the semantic and phonemic verbal fluencies. Due to our sample’s small size, we conducted a non-parametric Wilcoxon paired sample test for a group comparison.

### 2.6. Data Collection and Availability

Data were collected at the Hospital Center in Digne-les-Bains; the assessments were performed remotely and face to face by clinicians from the Memory Clinic in Nice. For remote assessments, the videoconference software recorded scores among other data such as videos or speech. These data were stored on a dedicated secured server, complying with healthcare data hosting regulations. They are not analyzed in this paper.

## 3. Results

33 Females and 17 Males participants aged between 40 to 86 years old were included (see Table 2).

Wilcoxon Paired Sample showed that the total MMSE scores, the LEXIS scores, the PVF and Praxis scores differed significantly between the FTF and TNP assessments (see Table 3). The MMSE, LEXIS and Praxis total scores were significantly higher in the FTF assessment. For the FCSRT, the total recall, delayed recall and recognition scores did not differ significantly. STROOP Color and Interference mean duration did not differ significantly. However, subjects were significantly faster in the Reading modality in the TNP assessment compared to the FTF. The SVF scores did not differ between both assessments.

Intraclass Coefficient Correlations were significant between both assessments for most test scores except for the recognition score of the FCSRT and the z-score of the SVF (see Table 4). Significant ICC ranged between 0.269 and 0.862. Level of reliability using Koo and Li’s guidelines can be defined as follows: *“Values less than 0.5 are indicative of poor reliability, values between 0.5 and 0.75 indicate moderate reliability, values between 0.75 and 0.9 indicate good reliability, and values greater than 0.90 indicate excellent reliability.”* [21]. Using these guidelines, tests that showed a moderate to good reliability were the Lexis and the Stroop (color and interference).

Results from the Acceptability Evaluation Scale are presented in this section. Mean scores of the usability experience rating and related comments are presented in Table 5. Almost all 50 subjects answered the first five questions. Question 1 to 4 were all above 6 (out of 7). Question 5′s mean score, related to the likelihood of recommending the assessment method, was above 9 (out of 10). 

Subjects’ answers to open questions of the Acceptability Evaluation Scales are presented in Table 6. 23 subjects answered Question 8. Comments can be divided into three categories: “Technical issues”, “Lack of human presence” and “Other comments”. 19 subjects answered Question 9 related with what subject’ like or disliked the most from the VTC experience. 15 subjects answered Question 10 related to their view on how to improve the system.

Only two subjects considered withdrawing from the study. 7 subjects preferred the TNP assessment and 13 the FTF condition. 18 subjects either wrote “both” or checked both boxes.

## 4. Discussion

Our paper aimed to investigate the validity and acceptability of the use of a videoconference system tailored to TNP assessment for older adults suffering from a cognitive complaint. Overall, our dedicated system enabled to perform a tele-neuropsychological assessment covering different cognitive domains with reliable results compared to a classical face to face assessment.

The TNP was well accepted by subjects regarding its usability. The gathered comments underlined how technical issues such as sound problems and internet access interruptions must be prevented. Some comments referred to a lack of human presence and their need for eye contact and proximity with the interlocutor, which is consistent with the literature and our experiences [22]. One subject preferred the TNP system as it provided a more secure environment to speak up more freely. Although only two possible choices were available to the question “which evaluation method do you prefer?”, most subjects answered both.

ICC results showed that most cognitive tests displayed poor to moderate reliability but significant agreement between both conditions except for recognition score of the FCSRT and the semantic verbal fluency. Visual tests such as the naming task and the Stroop test showed moderate to good agreement. Tests requiring auditive or verbal modalities however showed only poor reliability. Further results showed significant differences in several cognitive tests (see more details in Table 3) between FTF and the TNP conditions. Presentation time might have differed between the FTF and remote administration which could explain the differences observed in the recognition or naming task. Similar studies have shown that visual based tests show differences in results in the TNP condition leading to poor discrimination between healthy controls and patients with neurocognitive disorders [5].

Regarding the verbal fluencies results, the difference observed between conditions were surprising, since these tests are only based on verbal instructions and thus, are not supposed to be affected by the TNP condition. A potential explanation could be that the differences were caused using alternative versions of the SVF and PVF. There was a major incongruency between the two versions regardless of the condition (category “animal”/“fruits”, letter “P”/”R”) even using the z-score. These results suggest that there is a need to validate the digitalized cognitive tests that require visual cues with a broader and greater sample as means can be different due to a different visualization.

Based on our results and the recent literature, we could consider the use of TNP for patients with mild to major neurocognitive disorder, but the face-to-face interaction must not be dismissed. Authors have raised an interesting way to use TNP for first screening to determine how urgent the patients’ condition is and complete the whole assessment physically later at a second stage [23]. Most similar studies carried out were located in a hospital or a clinic equipped with a computer [4]. However, due to the COVID 19 pandemic, a more favorable approach would be at patients’ homes. A recent study demonstrated that in-home TNP was sensitive enough to detect cognitive impairment in older adults [24]. In home TNP results seem not to differ significantly from the face-to-face assessment. Despite, this method might be limited in this population specifically since often a certain unfamiliarity with recent technology usage can be encountered. In turn, other authors found that this was actually the case for control groups only but not for subjects with neurocognitive disorders [24]. Additionally, the uncontrolled environment can add distractions and interruptions which can lead to invalidate the whole test [6]. Satellite clinics can be designed to perform TNP with proper high-speed internet and technological services such as web cameras, a large screen monitor etc.). This latter option can be more suitable and safer with appropriate sanitary measures.

It seems clear however that remote assessment technologies represent a great potential and will play a major role in future dementia research and particularly in clinical trials. It represents an opportunity for equity and to allow underrepresented population to get involved in research or simply access to health care services.

In regards to future perspectives, an active and strong implication of patients in the further development and design of such systems should be emphasized in order to increase a satisfactory TNP experience [3]. Additionally, other innovative approaches are needed to overcome the barrier of physical distance between patient and clinician and complete missing assessment aspects.

## 5. Limitations

Our study had two assigned assessors for each condition, two psychologists for the TNP in Nice and two psychologists for the FTF condition in Digne-les-Bains. Although the tests are standardized, it is well known that due to a lack of guidelines when administering the MMSE, differences can be observed between two assessors [24], regarding time allowed to answer in terms of seconds or minutes for instance. We did want to preserve ecological conditions for the way assessments are usually performed. The small sample size represents a limitation to guarantee that results can be generalized to the wider population.

## 6. Conclusions

The TNP platform shows good agreement with the classical neuropsychological assessment although differences were observed. The TNP platform is web-based and can therefore be used on any web browser whether on computers, tactile tablets, and mobile phones. Ideally, this new way of assessing must result in the creation of adapted norms since performances may vary/differ due to the administration method and the interface (smartphone, tablet, computer). Furthermore, it would be important to provide administration protocols adapted to remote testing, which include guidelines on how to ensure that the patient is following the tests protocols (e.g., orienting the camera so that the clinician can see the patient performing a paper-based test).

Based on the patients’ reports, some aspects of the interface would need improvements. For instance, the possibility to see the video of the clinician also during the neuropsychological testing could be considered to provide some visual feedback, eye contact, and the feeling to be in a dyadic interaction. However, this could also be distracting, so further studies are needed to clarify the added value of these kids of manipulations. Technical problems should be carefully considered, and remote neuropsychological assessments should be proposed only if the equipment is reliable and the internet connection stable. An unreliable neuropsychological assessment due to technical problems is hardly interpretable by clinicians.

The 2020 global pandemic highlighted several problems, particularly among the elderly who were the most vulnerable individuals. Memory centers had to significantly reduce the number of visits to protect this high-risk population. Individuals with a memory disorder or already treated by memory centers had to reschedule their appointments. These patients, especially in large cities, may already be waiting more than six months for an appointment. This is a worrying problem when we know that cognitive decline can progress quite rapidly. There is now a major challenge to detect, assess and manage while protecting patients and their clinicians.

## Figures and Tables

**Figure 1 diagnostics-12-00925-f001:**
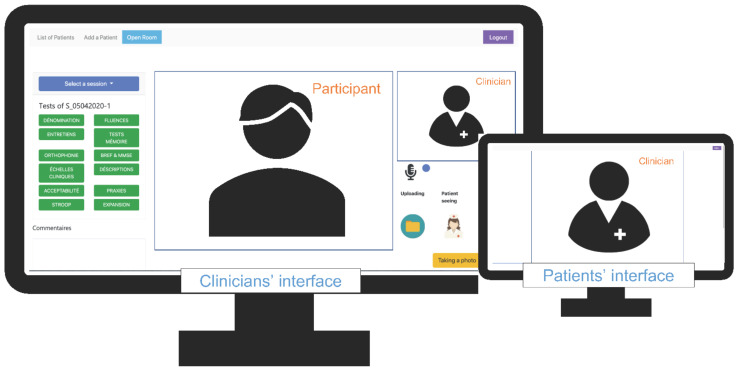
The teleneuropsychology platform developed for this research.

**Table 1 diagnostics-12-00925-t001:** Cognitive tests implemented.

Cognitive Functions	Cognitive Tests
Global cognitive functioning	MMSE [11]
Memory	Free and Cued Selective Reminding Test (FCSRT) [12]
Executive functions	STROOP test [13]
Language and Semantic Memory	Semantic and Phonological Verbal Fluency (SVF and PVF) [14]
	Naming task (Lexis) [15]
Praxis	Brief screening scale evaluating praxis abilities [16]

**Table 2 diagnostics-12-00925-t002:** Socio-demographic characteristics.

N	50			
**Sex**				
*Females, n* (*%*)	33	(66%)		
*Males, n* (*%*)	17	(34%)		
**Education**				
*Primary n* (*%*)	14	(28%)		
*Secondary, n* (*%*)	18	(36%)		
*High, n* (*%*)	18	(36%)		
			**Min**	**Max**
**Age, years** Mean (SD)	73.32	(9.89)	40	86
**Time delay between two assessments, in days** Mean (SD)	15.72	(3.43)	12	31
** *MMSE* ** **Mean (SD)**	28.24	(2.01)		

**Table 3 diagnostics-12-00925-t003:** Mean scores of the TNP and FTF assessments.

Cognitive Tests	N	TNP		FTF		*p* *
MMSE ^1^, total score	50	26.42	(3.57)	28.24	(2.01)	0.000
FCSRT ^2^						
Total recall score	42	43.91	(4.65)	42.91	(7.12)	0.872
Delayed recall	42	15.16	(1.67)	15.07	(1.88)	0.648
Recognition score	41	15.70	(1.52)	15.69	(1.02)	0.852
Lexis, total score	46	56.74	(5.71)	58.34	(5.56)	0.002
Stroop						
Color, duration (s)	42	77.14	(24.32)	72.62	(18.50)	0.127
Reading, duration (s)	42	54.10	(15.95)	59.22	(21.32)	0.021
Interference, duration (s)	40	153.44	(53.83)	152.52	(61.30)	0.132
SVF (z-score)	47	−0.33	(1.28)	0.34	(3.34)	0.256
PVF (z-score)	48	0.00	(1.15)	0.36	(1.02)	0.005
Praxis Total score	50	19.96	(2.32)	22.24	(1.45)	0.000

* Wilcoxon paired sample ^1^ MMSE: Mini Mental State Examination, ^2^ FCSRT: Free Cued Selective Reminding Task, Rappel Libre Rappel Indicé 16 items. Mean (Standard Deviation).

**Table 4 diagnostics-12-00925-t004:** Intraclass Coefficient Correlation results between FTF and TNP cognitive test scores.

Cognitive Tests		ICC			
		Coefficient	Lower Bound	Upper Bound	*p*
MMSE	Total Score	0.371	0.080	0.598	0.001
FCSRT	Total recall score	0.487	0.216	0.688	0.001
	Delayed recall	0.269	−0.040	0.529	0.043
	Recognition score	−0.048	−0.357	0.266	0.615
Lexis	Total score	0.862 **	0.715	0.929	0.000
STROOP	Color, duration (s)	0.569 *	0.327	0.741	0.000
	Reading, duration(s)	0.439	0.163	0.652	0.002
	Interference, duration (s)	0.643 *	0.421	0.793	0.000
Verbal fluency	Semantic (z-score)	0.084	−0.199	0.356	0.283
	Phonological (z-score)	0.445	0.192	0.644	0.000
Praxis	Total score	0.335	−0.093	0.643	0.000

* Moderate level of reliability (0.5–0.75); ** good level of reliability.

**Table 5 diagnostics-12-00925-t005:** Mean scores of the Acceptability Evaluation.

	N	Mean	(SD)
Q1. Globally, I’m satisfied with this experience *“I thought it would be more difficult for me”* *“The sound should be adjusted to avoid echo, reverberation”*	50	6.56	(0.84)
Q2. Globally, the system was easy to use *“childish”, “concentration was more difficult for me”* *“Yes, the system is easy to use”*	50	6.5	(0.71)
Q3. Instructions were clear *“But not clear enough because of the sound”* *“Very clear and in an accessible language.”*	50	6.71	(0.59)
Q4. I would repeat this experience in the future *“Yes, if it is necessary”* *“Without any problem or apprehension”*	47	6.53	(1.06)
Q5. On a scale from 1 to 10, how likely would I recommend this assessment method?	48	9.06	(1.67)

**Table 6 diagnostics-12-00925-t006:** Answers from the subjects’ experience of the TNP. (Translated from French to English).

Answers from the Subjects’ Experience of the TNP
Q8. What was missing or disappointing during your experience? (*n* = 23) **Technical Issues** “*Sound problem*”—“*computer bug”—“computer stress”—“computer shut down to perform updates”* **Lack of human presence** *“Drink water”—“human presence*” **Other comments** *“Nothing” (*n* = 10)—“*no *rather pleasantly surprised puts a distance which is rather facilitating”—« very good experience »“”*
Q9. What did you like most/least about this process? (*n* = 19) **Like** “*I felt free in front of the computer (gestures or body postures “for me” to focus)”* *“The kindness of the people that took care of me”* *“Real size of the clinician’s face*” **Dislike** *“Quite difficult”—“Lack of proximity to the interlocutor”* *“Gaze, I felt less presence”*
Q10. What would be the way to improve the system? (*n* = 15) *“To systematize it through the mutual insurance companies”* *“Computer placement (screen) hurts the eye”* *”Very good internet connectivity”*

## Data Availability

The data that support the findings of this study are available from the corresponding author upon reasonable request.

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
