# Peer review of "Feasibility Study of an Internet-Based Platform for Tele-Neuropsychological Assessment of Elderly in Remote Areas"

_diagnostics, 2022, doi:10.3390/diagnostics12040925_

Round 1

Reviewer 1 Report

The manuscript proposed by Zeghari and colleagues reports a crossover study aimed at verifying whether the remote assessment is reliable compared to classical testing. 50 patients with cognitive complaints were recruited and tested both face-to-face and via a Teleneuropsychology (TNP) system. The results show that: 1. for some tests, face-to-face patients' performance is better than that obtained with the TNP system; 2. Intraclass Coefficient Correlations were significant between both assessments for most test scores; 3. The teleneuropsychology system has been well accepted by participants.

Overall, the authors conclude that the TNP platform showed good agreement with the classical neuropsychological and was well accepted by subjects regarding its usability.

The manuscript is well organized and written, the aims are clear, the methodology quite well defined and developed, the results clearly presented and well discussed.

I have only some minor comments:

  1. It is not clear in the Methods section where the patients were during TNP assessment (at home or at the hospital) and which kind of device they used? Please be more clear about this point.
  2. In the Data Analysis section, the Authors said “We compared, in the digit span tests, the greatest number series recalled both forwards and backward” (lines 149 and 150) but previous in the Procedure/Protocol section the Authors said that Digit Span Onwards and Backwards, with some other neuropsychological tests, is not included in the analysis since it was not performed face-to-face (lines 117-119). Please verify.
  3. In the Procedure/Protocol section, the Authors said that “A comprehensive neuropsychological assessment was administered face-to-face (FTF) and via the TNP system” (lines 106 and 107) and refer to Table 1. The presented neuropsychological battery is far to be a comprehensive one. Executive functions are complex and they can't be assessed just with the Stroop test. Moreover, several other cognitive functions are not assessed (e.g. Visuospatial functions, Speed of processing, Selective attention, and so forth). I think that the proposed battery could be conceived as a short battery and should not be presented as comprehensive.
  4. In the Results section would be useful for the readers to have a table with demographic and clinical characteristics of the participants (e.g. age, sex, education, MMSE, autonomy in daily life, and so on).
  5. Would be interesting an interpretation and discussion of the Intraclass Coefficient Correlation results according to Koo and Li’s Guideline (Koo, T. K., & Li, M. Y. (2016). A guideline of selecting and reporting intraclass correlation coefficients for reliability research. Journal of chiropractic medicine, 15(2), 155-163).

Author Response

  1. It is not clear in the Methods section where the patients were during TNP assessment (at home or at the hospital) and which kind of device they used? Please be more clear about this point. 

We thank the Reviewer for the positive evaluation of our manuscript. Patients were received at the hospital of Dignes-les-bains, and performed the assessment on a computer. A nurse assisted them in establishing the connection to the platform. We added now added more information on these aspects in  the section “2.2 Procedure/Protocole” (see paragraph 2 ).

2. In the Data Analysis section, the Authors said “We compared, in the digit span tests, the greatest number series recalled both forwards and backward” (lines 149 and 150) but previous in the Procedure/Protocol section the Authors said that Digit Span Onwards and Backwards, with some other neuropsychological tests, is not included in the analysis since it was not performed face-to-face (lines 117-119). Please verify.

Thank you for your comment. There was a mistake for the data analysis section. We removed the digit span task information and added what we actually compared (detailed scores of each test) (see section 2.5. data analysis paragraph 2). 

3. In the Procedure/Protocol section, the Authors said that “A comprehensive neuropsychological assessment was administered face-to-face (FTF) and via the TNP system” (lines 106 and 107) and refer to Table 1. The presented neuropsychological battery is far to be a comprehensive one. Executive functions are complex and they can't be assessed just with the Stroop test. Moreover, several other cognitive functions are not assessed (e.g. Visuospatial functions, Speed of processing, Selective attention, and so forth). I think that the proposed battery could be conceived as a short battery and should not be presented as comprehensive.

The reviewer is right. The battery was selected to match both memory centers' usual neuropsychological assessment. However, many tests were either removed or impossible to perform (visuospatial functions are usually assessed using physical objects). We now removed the misleading words “comprehensive” and “complete”.

4. In the Results section would be useful for the readers to have a table with demographic and clinical characteristics of the participants (e.g. age, sex, education, MMSE, autonomy in daily life, and so on). 

We thank the reviewer for this suggestion. We have now added a table 2 with socio-demographics in the result section. 

5. Would be interesting an interpretation and discussion of the Intraclass Coefficient Correlation results according to Koo and Li’s Guideline (Koo, T. K., & Li, M. Y. (2016). A guideline of selecting and reporting intraclass correlation coefficients for reliability research. Journal of chiropractic medicine, 15(2), 155-163).

We thank the reviewer for the advice. We added the sentence : “ICC estimates and their 95% confidence intervals were calculated using SPSS statistical package version 23 (SPSS Inc, Chicago, IL) based on an absolute-agreement, single measure, 2-way mixed-effects model.” as suggested by the authors, which gives information on the model we used (section 2.5. data analysis paragraph 1 ). 

We interpreted the agreement in the discussion using Koo and Li’s Guidelines as suggested by the reviewer. 

We modified this sentence in the discussion : "ICC results showed that most cognitive tests displayed poor to moderate reliability but significant agreement between both conditions except for recognition score of the FCSRT and the semantic verbal fluency. Visual tests such as the naming task and the Stroop test showed moderate to good agreement." (discussion paragraph 3)

Reviewer 2 Report

a new online solution for cognitive evaluation of the eldery is assessed in this study intended to be used as a telemedicine tool. It can be improved in writing style where for example one sentence was written as a paragraph. It can also be improved by discussing the perspective of the tool and what are the potential solutions to the issues raised by the users. I think of more areas to discuss that were missing for example how clinicians could make sure that the patient follows test protocols as they would do in clinics. Another issue is that the tool is claimed to be a "complete neuropsychological assessment", I think the word complete here may be too strong for an assessment tool as it might be argued that some other tests are missing. Some terms are not defined: SVF and PVF. The age range (40 to 86 years old) is too wide to be defined as elderly, the authors should stratify the subjects to two age groups of middle aged and the elderly and see what changes it makes to the results. 

Author Response

We thank the reviewer for their comment. We answered each comment separately. 

  • It can be improved in writing style where for example one sentence was written as a paragraph. 

As suggested we tried to shorten several sentences to improve readability

  • It can also be improved by discussing the perspective of the tool and what are the potential solutions to the issues raised by the users.

Thanks for the suggestions. We now enriched the conclusion session. Based on the patients’ reports, some aspects of the interface would need improvements. For instance, the possibility to see the video of the clinician also during the neuropsychological testing could be considered to provide some visual feedback, eye contact, and the feeling of being in a dyadic interaction. However, this could also be distracting, so further studies are needed to clarify the added value of these kinds of manipulations. Technical problems should be carefully considered, and remote neuropsychological assessments should be proposed only if the equipment is reliable and the internet connection stable. An unreliable neuropsychological assessment due to technical problems is hardly interpretable by clinicians.

  •  I think of more areas to discuss that were missing for example how clinicians could make sure that the patient follows test protocols as they would do in clinics. 

Thanks for the interesting suggestion. We now added to the conclusions that it would be important to provide administration protocols adapted to remote testing, which include guidelines on how to ensure that the patient is following the tests protocols (e.g., orienting the camera so that the clinician can see the patient performing a paper-based test) 

  • Another issue is that the tool is claimed to be a "complete neuropsychological assessment", I think the word complete here may be too strong for an assessment tool as it might be argued that some other tests are missing. 

We agree with the reviewers’ comment and replaced the word ‘complete’ by ‘comprehensive’.

  • Some terms are not defined: SVF and PVF.

Thanks, Semantic and Phonological Verbal Fluencies' acronyms are now defined in the manuscript. 

  • The age range (40 to 86 years old) is too wide to be defined as elderly, the authors should stratify the subjects to two age groups of middle aged and the elderly and see what changes it makes to the results. 

Thanks for this comment. Only one subject was 40 years old and displayed memory and attention problems. We now reported the participants demographics in Table 2.

Reviewer 3 Report

It is worth emphasizing what the purpose of the diagnosis with the use
of the described TNp system is. Is it a screening diagnosis?

Did the Authors control for the type of device on which the
subjects performed the tests (phone, tablet, computer)? and did they
require the help of third parties?

Please ensure that someone independent and highly proficient in written English has thoroughly checked/edited your manuscript (for example line 209).

Author Response

  • It is worth emphasizing what the purpose of the diagnosis with the use of the described TNp system is. Is it a screening diagnosis? 

We thank the reviewer for his question. The tool is designed for assessment and can be used. It could be used to detect at early stages neurocognitive symptoms. 

  • Did the Authors control for the type of device on which the subjects performed the tests (phone, tablet, computer)? and did they require the help of third parties?

Patients all performed the assessment on a desktop computer in a quiet room within the hospital of Digne-les-bains.  A nurse from the hospital switched the computer on, and logged participants to their session. She also provided material for the tests (white paper and a pencil for the MMSE test). When the participants were installed and the Psychologist connected to the session as well, the nurse left the room but stayed in another adjacent room in case of any technical issues or if the participant was in need of anything (water, discomfort, volume adjustment). 

We specified this in the methodology section “2.2 Procedure/protocol”. 

  • Please ensure that someone independent and highly proficient in written English has thoroughly checked/edited your manuscript (for example line 209).

Thanks, the manuscript was proofread.

Round 2

Reviewer 2 Report

The authors have improved the manuscript to address some issues and moderate the conclusion based on the findings. I did not find the authors discuss the broad age range they used (beginning from 40) as the main application seems to be targeted for the elderly. Thus it is needed to discuss that point further or mention it as a limitation.